# Cross-Sectional Study Assessing Management Practices and Udder Health in California Sheep Flocks and Seroprevalence of Small Ruminant Lentivirus

**DOI:** 10.3390/ani14162332

**Published:** 2024-08-13

**Authors:** Rose Digianantonio, Alda F. A. Pires, Roselle Busch

**Affiliations:** Department of Population Health and Reproduction, School of Veterinary Medicine, University of California, Davis, CA 95616, USAapires@ucdavis.edu (A.F.A.P.)

**Keywords:** mastitis, ovine progressive pneumonia (OPP), small ruminant lentivirus (SRLV), sheep, California, multiple correspondence analysis (MCA)

## Abstract

**Simple Summary:**

California was the second largest sheep-producing state in the US, with an inventory of 533,000 head of sheep and lambs in 2022. Mastitis is a health and welfare concern for ewes and lambs. Mastitis is one of the top reasons for the removal of ewes from the flock and antibiotic use in ewes. In this study, California sheep producers were surveyed on the management of ewes. Seventy-one surveys were completed. Most respondents had flocks of ≥100 sheep (54%; 38/71). Twenty-three percent (16/70) of the respondents reported having >5% udder abnormalities in a flock per lactation. Further analysis showed two clusters among participating farms. Larger flock sizes, the use of breeding ewes for meat or wool production or contract grazing, and more extensive management practices were associated with >5% udder abnormalities and ≥5% orphan lambs. A subset of the respondents (20) participated in serological testing for small ruminant lentivirus (SRLV) in sheep. A total of 1106 ewes or ewe lambs were sampled. The ewe-level seroprevalence was 14.1% (183/1106), and the flock-level seroprevalence was 75% (15/20) of the sampled farms. Our results describe flock demographics, management practices, and producer-reported udder health abnormalities in California ewes. Larger flock sizes and extensive management practices could be risk factors for udder abnormalities in ewes.

**Abstract:**

(1) Background: Information is lacking on small ruminant lentivirus (SRLV) status, prevalence, risk factors, and control measures for mastitis in California ewes. The goal of this survey was to outline characteristics of the sheep industry in California related to udder health and mastitis management. (2) Methods: An online survey consisting of 48 questions was completed by respondents between April 2022 and February 2023. Descriptive analysis and chi-squared tests were conducted to evaluate associations between variables. A multiple correspondence analysis (MCA) of general management practices, udder health management, and flock demographics was performed to assess clustering. A subset of respondents (20) participated in SRLV serology testing. (3) Results: Seventy-one completed surveys were submitted. The MCA showed two clusters. Larger flock sizes, the use of breeding ewes for meat or wool production or contract grazing, and extensive management practices were more closely related to >5% udder abnormalities per lactation and ≥5% orphan lambs. The flock-level seroprevalence of SRLV was 75% (15/20), and ewe-level seroprevalence was 14.1% (183/1106). (4) Conclusions: The results of this study highlight areas that need further research, such as exploring differences in mastitis and SRLV incidences among management systems, the efficacy of mastitis treatments, and education on critical timepoints for mastitis diagnosis and control.

## 1. Introduction

General information is lacking on the occurrence, risk factors, economic impact, and control measures for mastitis in sheep in California. According to the National Agricultural Statistics Service’s 2022 Census of Agriculture, California was the second largest sheep producing state in the United States, with an estimated 533,000 head of sheep in 2022 [1]. About half of these sheep (270,000 head) were breeding ewes one year of age or older [1].

Mastitis is a health and welfare concern for breeding ewes and lambs in sheep production. The 2011 Sheep Study conducted by the USDA’s National Animal Health Monitoring System (NAHMS) indicated that mastitis was one of the top reasons for culling breeding ewes and the second most common reason for antibiotic use in sheep operations [2]. Additionally, producers indicated starvation was the second most common cause of lamb mortality [2]. Udder abnormalities play a significant role in lamb starvation when a ewe does not allow lambs to nurse or her milk production is insufficient [3]. The *2016 American Sheep Industry Association Report* stated that producers identified mastitis as one of the most common disease management challenges in breeding ewes. Among breeding-ewe diseases, mastitis was considered to have the second greatest economic impact on producers, and ovine progressive pneumonia (OPP) was considered to have the fourth greatest impact [4]. However, there is a lack of research describing management factors related to ewe mastitis and SRLV prevalence in the Western United States.

Ovine progressive pneumonia (OPP) is a clinical manifestation of small ruminant lentivirus (SRLV) in sheep (also known as maedi–visna virus). Veterinarians, animal scientists, and shepherds in the US commonly refer to all the clinical manifestations of SRLV in sheep as OPP; however, throughout this manuscript, SRLV will be used to indicate disease exposure or status, as OPP is not fully encompassing. Infected sheep are chronic carriers of the virus and may or may not show clinical signs. The virus is slow-growing and initiates an inflammatory lymphoproliferative response most commonly in the lungs, mammary tissue, joints, or central nervous system [5]. This disease manifests clinically as sheep with a low body condition score, pneumonia, and fibrotic udders (“hard bag”). Thus, SRLV-positive ewes may be less fertile and produce lambs with lower birth or weaning weights [6,7]. Decreased milk production may lead to an increased number of orphan lambs; however, this has not been clearly demonstrated.

The seroprevalence of SRLV in sheep is not well documented in California. The NAHM’s 2001 Sheep Study tested sheep across the United States for SRLV and found 21.6% of operations to have at least one SRLV-positive sheep and 17.8% of sheep to be SRLV positive in the Pacific region of the United States (California, Oregon, and Washington) [2]. A study completed in Wyoming found an apparent seroprevalence of SRLV of 18% at the ewe level and 47.5% at the flock level [8]. The same study reported that larger flocks (median size = 406.5 ewes) tend to have seropositive sheep more often than smaller flocks (median size = 257 ewes). Flocks that grazed on open ranges were at 3.4 times the odds for being OPP positive than flocks that grazed on fenced ranges or pastures. Studies in other countries have also reported larger flock sizes as a risk factor for SRLVs, as well as parity and more intensive management practices [9,10,11]. Although these studies can help to understand disease dynamics in flocks, sheep production systems and management vary greatly by country, and regional research is needed.

A few studies have investigated risk factors and etiologies for mastitis in range ewes in the west [12,13,14] as well as the effect of mastitis on lambs [15]. These studies were aimed at examining ewe-level risk factors. The impact of SRLV seroprevalence at the flock level and effect on production in lieu of other management practices have not been well described in California. The main objective of this study was to describe flock-level risk factors for udder abnormalities and characterize California sheep breeding–ewe management.

## 2. Materials and Methods

### 2.1. Participant Recruitment and Survey Administration

In this cross-sectional study, the study population consisted of an estimated 1571 current sheep operations in the state of California with breeding ewes [1]. An online survey was constructed in Qualtrics XM (Qualtrics, Provo, UT, USA), which consisted of 48 questions, including a combination of yes or no, categorical, multiple response, and open-ended questions (Appendix A). The survey was accessible online via a direct link or a quick-response code on information pamphlets (Appendix A) dispersed at regional sheep producer meetings, via electronic mail, or social media websites starting on 30 April 2022. Respondents were reached via the University of California Cooperative Extension social media pages and the California Woolgrowers Association membership meetings and e-mail list. Data were analyzed based on surveys that were greater than 70% complete and submitted between 30 April 2022 and 28 February 2023.

### 2.2. Survey Development

The goal of the survey was to describe factors affecting udder health and lamb survival in California sheep production. Seven main categories were detailed in the survey: flock demographics, breeding management, pregnancy and lambing managements, udder conformation considerations, mastitis management, and lamb care. Two researchers developed the survey; it was pre-tested on a group of five sheep producers who provided feedback, and edits were made accordingly. Questions related to each category were written based on previous literature on management and disease data on udder health and lamb survival. Answers to multiple choice or multiselect questions were adapted based on the researchers’ knowledge of the sheep industry in California, feedback from the pre-test group, and write-in answers from respondents. For a subset of questions, skip logic was used so that respondents only answered questions that related to their management practices. Each question was translated to a code book that was used for data analysis. Data verification was performed by spot checking all the survey responses to the original Qualtrics inputs by IP addresses. Nine surveys had less than 70% of the questions answered and were not used in the descriptive or MCA analysis. Surveys with missing responses to any of the 10 variables used in the MCA were excluded from that analysis.

### 2.3. Data Management and Descriptive Statistical Analysis

Data were exported from Qualtrics XM and imported into Microsoft Excel 2019 for descriptive statistical analysis. Responses to open-ended questions (n = 1) were categorized as needed and fill-in responses were re-categorized or entered as an additional response category as deemed appropriate. For multiselect questions (n = 11), variables were consolidated if appropriate. For example, producers were asked when ewes were moved to close-up paddocks/pastures/pens for observation prior to lambing. Possible responses included greater than 3 weeks before suspected lambing, 2–3 weeks before suspected lambing, 1 week before suspected lambing, and not moved prior to lambing (rangeland lambing). These were consolidated into responses “moved prior to lambing for observation” or “not moved prior to lambing for observation”. Dummy variables were created for multiselect questions, and each variable was re-classified as a binary response. For example, respondents were asked when they evaluated ewes for breeding soundness and were able to select all that applied among purchase, pre-breeding, lambing, weaning, or shearing. For breeding soundness, this resulted in five dummy variables being created with a yes or no response. Descriptive analysis and figures were completed using Microsoft Excel 2019 (Microsoft Corp., Redmond, WA, USA) and R Studio (PBC, Boston, MA, USA). Chi-squared tests of independence were performed to examine relationships between variables of interest, including the flock size, production purpose, use of lambing jugs, ewes moved to close-up pens prior to lambing, percentage of udder abnormalities, percentage of orphan lambs, percentage culled for udder abnormalities, age of lambs at weaning, and SRLV seroprevalence. A map of the distribution of the survey responses was created using ArcGIS Pro (ESRI, Redland, CA, USA).

### 2.4. Multiple Correspondence Analysis

Multiple correspondence analysis (MCA) was performed using R Studio for Desktop Version 2022.7.2.576 and the “FactoMineR” package (PBC, Boston, MA, USA). Ten variables were selected for analysis, including the percentage of udder abnormalities per lactation (less than or equal to 5% or greater than 5% per lactation), percentage of orphan lambs (less than 5% or greater than or equal to 5%), purpose (meat, wool, or contract grazing vs. club or seedstock), flock size (fewer than 100, 100–499, or 500 or greater), selection criteria for keeping ewes (none, select for production ability, select for udder characteristics, or select for both), lambing jug use (yes or no), close-up management (moved prior to lambing or not moved prior to lambing), weaning age (less than 90 or greater than or equal to 90 days old), mastitis treatment with antibiotics (yes or no), and culled for mastitis (yes or no). These variables were selected for the MCA because previous literature described their associations with mastitis; they represent substantial management differences, or they are directly related to lactation. Of the 71 completed surveys, 6 surveys could not be used in the MCA, as they were missing values in one or more of the included variables.

### 2.5. SRLV Testing

A subset of survey respondents, with at least 30 breeding ewes, volunteered to participate in SRLV testing of their flocks. Blood was collected by one of the researchers, the flock veterinarian, or veterinary technical staff via venipuncture of the jugular vein with a vacutainer red top tube. Female sheep 12 months of age and older were eligible for SRLV testing. The number of sheep sampled per flock was based on an assumed 22% flock prevalence and an 18% within-flock prevalence of SRLV, a test sensitivity of 95.5%, and a specificity of 100% [2]. To achieve 95% confidence of the estimated prevalence, all the eligible sheep were sampled from smaller flocks (<50), and at least 60 ewes were sampled from larger flocks (>500). Animals were randomly selected chute side for larger flocks. Samples were placed on ice and transported to a research laboratory at the University of California, Davis. Samples were centrifuged at 1500× *g* for 10 min for serum separation. A commercially available Small Ruminant Lentivirus Antibody Test Kit (cELISA) was used to analyze the serum for SRLV antibodies (VMRD, Pullman, WA, USA). A positive result was defined as >20.9 percent inhibition [16].

## 3. Results

### 3.1. Response Rate

The survey link was e-mailed to 403 producers, with a response rate of 19.8% (80/403). There were 80 respondents to the survey, but nine were not included in the analysis, as less than 70% of the survey was completed. Thus, 71 completed surveys were analyzed. Percentages are rounded to the closest whole number for reporting values.

### 3.2. Descriptive Statistics

#### 3.2.1. Demographics

See Figure 1 for the distribution of the survey respondents by region and flock size. The majority of the respondents were from the northern (34%, 24/71) and Sierra regions (31%, 22/71) of California. The flock size varied, with 46% (33/71) of the respondents having flocks smaller than 100 sheep, 24% (17/71) of the operations having 100–499 sheep, and 30% (21/71) having flocks of greater than or equal to 500 sheep (Table 1). Most operations had more than one production purpose (65%, 37/57). Therefore, the purpose was grouped to represent operation types that are closely related. Sheep used for meat, wool, and contract grazing were grouped together and represented 51% (36/71) of the respondents. Club or seed stock-breeding operations represented 49% (35/71) of the respondents (Table 1).

#### 3.2.2. Breeding Management

From April to July was the most common breeding season for 44% (31/70) of the respondents, followed by from July to September (27%, 19/70), from October to January (24%, 17/70), and from January to March (4%, 3/70) (Table 1). In reference to biosecurity practices, 15% (11/71) of the operations indicated they routinely tested breeding ewes for SRLV. Just over half of the respondents purchased replacement ewes (52%, 37/71), meaning they would intermittently have new additions to their operation. Of those respondents, 5% (2/37) tested the replacements for SRLV.

Ewes were evaluated for breeding soundness (the ability to become pregnant and raise a lamb) at pre-breeding (63%, 45/71) and weaning (62%, 44/71) most commonly. Most respondents considered the general conformation (93%, 66/71) and mothering ability (82%, 58/71) and bred in the desired breeding window (58%, 41/71) when keeping a ewe in the breeding flock. For the subset of producers who purchased replacement ewes, the most commonly used selection criteria were the general conformation (92%, 33/36), reputation of the seller (89%, 32/36), and mothering ability (56%, 20/36) (Table 1). The udder conformation was considered by 56% (40/71) of the respondents for keeping ewes and 28% (10/36) of the respondents for selecting replacements.

#### 3.2.3. Udder Health

Respondents reported that flock owners were most commonly the ones examining udders for conformation or signs of disease (81%, 57/70), and udder abnormalities were noted the most often at lambing (66%, 47/71) and less often around mid-lactation (41%, 29/71) and weaning (35%, 25/71). The most common abnormalities for the participants not keeping a ewe for breeding were hard bag (66%, 47/71), a history or the presence of mastitis (54%, 38/71), and lumps in the udder (38%, 27/71) (Table 1).

Respondents reported mastitis more often from udder palpation abnormalities (92%, 66/71) rather than lamb performance or milk abnormalities. Most respondents used veterinarians (79%, 56/71) or other producers (48%, 34/71) for advice about treating mastitis. Injectable antibiotics were used for the treatment of the mastitis by 77% (55/7) of the respondents, while 67% of the respondents used intramammary antibiotics (26/71). Non-steroidal anti-inflammatories (NSAIDS) were used as a part of mastitis treatment protocols by 15% of the respondents (11/71). Culling was also a common production decision for ewes with mastitis for 52% of the respondents (37/71) (Table 1).

Producers were asked to report the percentage of udder abnormalities noted per lactation (<1%, 1–5%, 6–10%, 11–20%, or >20% udder abnormalities). Most respondents (77%, 54/70) reported having ≤5% of ewes with udder abnormalities, and 23% (16/70) reported having >5% with udder abnormalities per lactation.

#### 3.2.4. Lambing Management

Most respondents moved ewes into close-up pens before their expected lambing date (77%, 54/70). Lambing jugs, small pens used to house ewes and lambs at and after lambing, were commonly used by respondents (76%, 53/70). Of those who used lambing jugs, most moved ewes and lambs into jugs within 12 h after lambing (51%, 27/53) or when the ewe started showing signs of labor (30%, 16/53). Others used lambing jugs only for weak lambs and maternal behavioral issues (19%, 10/53). Straw (74%, 39/53) and metal paneling (66%, 35/53) were the most common materials used in lambing jugs. About half of the respondents who used lambing jugs moved pairs from jugs 2–3 days after lambing (53%, 26/49), 35% (17/49) moved pairs 1 day or less after lambing, and 12% (6/49) moved pairs greater than 3 days after lambing. Most respondents changed jug bedding between ewes (73%, 38/52) or daily (17%, 9/52). Ewes and lambs not moved to jugs, lambed on fenced pasture (52%, 29/56) or in an open barn or shed among other ewes (30%, 17/56) (Table 1).

Lamb mortality was reported to be the most common in neonates between 0 and 5 days old (91%, 62/68). Lamb processing (e.g., ear tag placement, castration, and tail docking) occurred within 1 week of age for 49% (34/69) of the respondents or between 1 and 4 weeks old (39%, 27/69). Most respondents weaned lambs at 90 days old or older (66%, 45/68) and the rest partook in early weaning (34%, 23/68). Most respondents reported having less than 5% of orphan lambs per lambing season (80%, 55/69).

### 3.3. Chi-Squared Analysis

There was a significant relationship between the flock size and purpose. Smaller flocks (<100 sheep) were more likely to have breeding sheep for club and seedstock purposes (70%, 23/33), while large flocks (≥500 sheep) were more often meat, wool, or contract-grazing flocks (90%, 19/21; *p*-value < 0.001). Additionally, larger flocks were more likely to be culled for mastitis than medium and small flocks (76%, 16/21 versus 42%, 21/50; *p*-value = 0.03) and weaned at an older age (≥90 days old) (95%, 18/19 versus 45%, 22/49; *p*-value < 0.001). Respondents who weaned lambs at ≥90 days old were more likely to breed sheep for meat and wool production (69%, 31/45 versus 9%, 2/23; *p*-value < 0.001), not move ewes prior to lambing (32%, 14/44 versus 4%, 1/23; *p*-value = 0.02), and not use lambing jugs (33%, 15/45 versus 4%, 1/23; *p*-value = 0.02) compared to those who weaned lambs at less than 90 days old.

A higher percentage of udder abnormalities was associated with ≥5% orphan lambs (50%, 7/14 versus 15%, 8/53; *p*-value = 0.01) and the use of culling as a management decision for mastitis (35%, 13/37 versus 9%, 3/33; *p*-value = 0.02).

### 3.4. Multiple Correspondence Analysis

Ten variables were selected for analysis, and categories were collapsed as appropriate for interpretation. The selected variables were the percentage of udder abnormalities per lactation (less than or equal to 5% or greater than 5% per lactation), percentage of orphan lambs (less than 5% or greater than or equal to 5%), purpose (meat, wool, or contract grazing vs. club or seedstock), flock size (fewer than 100, 100–499, or 500 or greater), selection criteria for keeping ewes (none, selected for production ability, selected for udder characteristics, or selected for both), lambing jug use (yes or no), close-up management (moved prior to lambing or not moved prior to lambing), timing of weaning (less than 90 or greater than or equal to 90 days old), treated mastitis with antibiotics (yes or no), and culled for mastitis (yes or no). Figure 2 visually depicts the MCA results. The first and second dimensions constitute 34.9% of the explained variation in the dataset. The color scale represents the contributions of the categorized different responses of each variable to the variation in the data. Responses that are light yellow contributed the least to the variation in the survey data, and responses that are dark blue contributed the most (as indicated in figure legend). Two clusters are present in the MCA. The cluster in the second and third quadrants (outlined in gold) includes flocks smaller than 500 sheep, breeding ewes used for club and seedstock production, and responses representative of intensive flock management (the use of lambing jugs, moving ewes prior to lambing, weaning lambs younger, and treating mastitis with antibiotics). Additionally, this cluster represented respondents who focused their selection criteria for keeping ewes in the breeding flock on the production ability or overall production and udder conformation of the ewe. These flock demographics and management characteristics were more closely associated with lower udder abnormalities noted in ewes per lactation and a lower orphan lamb percentage. Contrarily, the cluster in the first and fourth quadrants (outlined in navy) includes large flocks, breeding ewes used for meat and wool production or contract grazing, and responses representative of extensive flock management (did not use lambing jugs, did not move ewes prior to lambing, weaned older lambs, did not treat mastitis with antibiotics, and culled ewes with mastitis). This cluster represented respondents who do select ewes based on udder characteristics or do not use any selection criteria for keeping breeding ewes. These flock demographics and management characteristics were more closely associated with higher udder abnormalities noted in ewes per lactation and a higher orphan lamb percentage.

### 3.5. SRLV Testing

Of the 71 surveys used in the final analysis, 20 respondents enrolled in SRLV seroprevalence testing (28%). Of these 20 respondents, 11 had 500 or greater sheep, and nine had fewer than 500 sheep. Only six were seedstock or club lamb operations, and 14 were primarily meat, wool, and/or contract-grazing flocks. On average, 55 ewes or ewe lambs were sampled per flock, with a total of 1106 blood samples collected. The ewe-level OPP seroprevalence was 16.5% (183/1106), and the flock-level seroprevalence was 75% (15/20). Flocks were considered to be positive for SRLV if they had at least one ewe test positive. Among the positive flocks, the ewe-level seroprevalence ranged from 1.6% to 56.4%.

To evaluate risk factors for SRLV, flocks were divided into three categories: flocks with an SRLV seroprevalence of 0%, 1–10%, or >10%. Of the 20 respondents who completed SRLV testing, 25% (5/20) had no SRLV seropositive ewes, 30% (6/20) had 1–10% seroprevalence, and 45% (9/20) had >10% seroprevalence. Higher SRLV seroprevalences (>10%) were seen more frequently in flocks of 500 or greater sheep compared to small and medium-sized flocks (64% (7/11) versus 22% (2/9); *p*-value = 0.003). Flocks for the purpose of club or seedstock production were more likely to have no ewes test positive for SRLV (57% (4/7) versus 8% (1/13); *p*-value = 0.05).

## 4. Discussion

### 4.1. Survey Limitations and Biases

In this cross-sectional study, sheep producers were surveyed based on flock demographics and breeding-ewe management. The 19.8% response rate was lower than similar studies of the sheep industry in California [17]. The lower response rate could have led to sampling bias in the data analysis. Specifically, the dispersal of the survey via electronic means only, limits respondents to those with access to the necessary technology. Surveys with multimodal dispersal methods improve response rates and limit sampling and response biases [18,19]. Additionally, the primary method for notifying producers was through the California Woolgrowers Association. Members of a producer association may be more up to date on industry evolution or more established producers. These producers could have management practices that vary from those of the general population of producers in the state.

Larger flocks were over-represented in our survey results and could present sampling bias in the results. Flocks with greater than 100 sheep made up 54% of our respondents, and in the most recent census of the California sheep industry, only 16% of the breeding flocks in the state had greater than 100 sheep [1]. This over-representation of large flocks could be because of mastitis being perceived as a more prevalent issue in larger flocks, as was noted in the *2016 American Sheep Industry Priorities Report* [4]. Thus, larger-scale producers may have had a greater interest in completing the survey and SRLV testing than smaller-scale producers. Additionally, larger producers with commercial flocks are more likely to be a part of producer groups, such as the California Woolgrower’s Association, which is how participants were contacted to partake in the survey. The over-representation of larger flocks may indicate that the results are not generalizable to California sheep flocks. This could have been limited by performing outreach for survey participation to more diverse groups (e.g., 4H groups and small-farm flock producers). However, the participation of larger flocks means that more sheep were represented in the survey results. According to the most recent Census of Agriculture, larger flocks represent 90% of California’s sheep inventory [1].

### 4.2. Descriptive Results

Previous literature has reported an average incidence of 7% clinical mastitis in non-dairy sheep, ranging up to 20–30% incidence [6,12,20]. We surveyed producers on the perceived percentage of udder abnormalities per lactation not the number of confirmed mastitis cases per year. This was to encompass any udder issues that may lead to decreased milk production and lamb survival, such as previous evidence of mastitis and udder injuries. This is an indirect measure of mastitis incidence, so producers were asked to give a range of occurrence rather than a distinct number. Recall bias, information bias, and poor or inconsistent recordkeeping are factors that could affect a reported perceived percentage. For instance, producers reporting sheep with mastitis issues may be more likely to remember an accurate percentage of udder abnormalities in their flock compared to other producers. Bias in the outcome variable, the perceived percentage of udder abnormalities, is a limitation of this study.

Udder abnormalities were noted the most often at lambing, and palpation was the most common method of diagnosis. Noting abnormalities at lambing is an expected finding because of increased observations, the opening of the teat canal, and periparturient immunosuppression at that timepoint [15,21,22]. A part of the purpose of this survey was to recognize critical timepoints that would be important for sampling ewes for mastitis per owner perception and training producers to monitor for mastitis. From our results, lambing would be the most common timepoint for the recognition of clinical mastitis, followed by mid-lactation and weaning. These critical timepoints are similar to those for other species [23,24]. Previous studies in sheep have shown that clinical and subclinical mastitis cases are the most common around lambing (within 2 weeks), with secondary peaks around mid-lactation (30–45 days post lambing) or weaning [12,13,21,23]. Encouraging the use of other methods of diagnosis, such as the California Mastitis Test, lamb behavioral observations, and lamb weight gains at these timepoints is another valuable training opportunity for early diagnosis. The use of these methods may be helpful in determining appropriate treatment methods and preventing chronic changes to the udder. Overall, this could help producers to target mastitis diagnosis and treatment and prevent long-term udder issues in their flocks.

When antibiotics were used, injectable antibiotics were most commonly a part of the mastitis treatment plan and are the most indicated in animals with evidence of systemic infection due to mastitis. Thus, it would be important to understand if producers modify their treatment protocols based on the severity of the disease, ease of treatment, or perceived efficacy over other treatment methods, such as intramammary antibiotics. Further research is needed on the efficacy of antibiotic treatments and appropriate protocols for different types of producers. This would allow for the promotion of antimicrobial stewardship and use of targeted therapies.

### 4.3. Chi-Squared and Multiple Correspondence Analyses

The chi-squared analysis indicated a significant association between >5% udder abnormalities and ≥5% orphan lambs and vice versa. This association between perceived udder abnormalities and orphan lambs is expected because of decreased or abnormal milk production from the dam or udder pain when lambs attempt to nurse. The presence of this correlation in our data could indicate mastitis is associated with orphan lambs or that management practices can increase or decrease the occurrence of both. Orphan lambs lead to extra labor and financial costs for producers, and minimizing mastitis cases plays a substantial role in decreasing lamb morbidity and mortality [2,25,26]. Producers indicated that lamb mortality was the highest in neonates, which is consistent with previous research [27]. Mortality in neonates is commonly linked to the failure of passive transfer or starvation, often associated with udder issues in ewes [25,26,27]. This emphasizes the concern of mastitis in lamb welfare and survivability.

Few studies have examined flock-level risk factors for udder abnormalities in sheep. This MCA resulted in two clusters of respondents. Generally, one cluster represented smaller flocks, club or seedstock operations, more intensive management practices (the use lambing jugs, moving ewes prior to lambing, and practicing early weaning), and a lower percentage of perceived udder abnormalities and orphan lambs. Respondents in this cluster also tended to treat mastitis with antibiotics, were less likely to cull ewes for mastitis, and selected for production ability or both production and udder characteristics. The combination of these variables leads to possible conclusions that sheep in smaller, more intensive management systems are at a lower risk for developing udder issues and/or that udder issues are monitored and treated more efficiently. Additional research is necessary to determine if both conclusions play a role or one more than the other. Persson et al. (2021) found an association between flocks of greater than or equal to 160 ewes to have a higher rate of intramammary infections than small flocks [28]. However, the effect of intensive management on mastitis incidence is likely multifactorial and varies by the goals of a specific operation. The practice of early weaning has been thought to be associated with a higher risk of mastitis in ewes [29]. There was not a strong association between the weaning age and percentage of udder abnormalities noted on farms. However, the weaning age was associated with moving lambs prior to lambing and using lambing jugs. Knuth et al. (2022) examined subclinical mastitis prevalence at the ewe level in relation to lambing-jug bedding management and weaning management but did not look at the incidences between the use or non-use of jugs and the weaning time [14]. Further research is needed to determine the rates of mastitis in non-dairy sheep managed more intensively (using lambing jugs and weaning early) versus extensively.

### 4.4. SRLV Testing

Just over a quarter of the respondents participated in SRLV testing of their flocks. The 16.5% apparent prevalence in ewes is similar to previously reported values in US sheep [8,30]. However, an apparent flock prevalence of 75% is higher than the 47.5% and 36.4% prevalence found in the Wyoming-based and nationwide (NAHMS 2001 Sheep Study) studies performed, respectively. The Wyoming and NAHMS 2001 studies had a larger number of flocks sampled and sampled fewer ewes per flock. Sampling an increased number of ewes from each flock could have led to an increased likelihood of a positive detection. Additionally, respondents to the survey who were willing to participate in SRLV testing could have had an increased suspicion of SRLV related conditions in their flock, whereas the Wyoming and NAHMS 2001 studies had goals aside from SRLV testing.

Larger flocks and flocks for the purposes of meat and/or wool production(s) and contract grazing were more likely to have a higher seroprevalence of SRLV in ewes. These findings are in agreement with multiple studies on SRLVs in sheep and goats [2,8,9,10,11], including the 2001 NAHMS study, where the SRLV prevalence was greater in flocks of 500 or more ewes compared to smaller flocks. Larger flock sizes and commercial purposes were related in this study, which could explain why both were risk factors of increased SRLV seroprevalence. The tendency of flocks with higher SRLV seroprevalences to be treated for mastitis with antibiotics is an association that should be more closely examined.

## 5. Conclusions

Our results describe flock demographics, management practices, and producer-reported udder health abnormalities in California ewes. Larger flock sizes and extensive management practices could be risk factors for udder abnormalities in ewes. Further research on udder health in California sheep is needed, including differences in mastitis and SRLV incidences among management systems, reasons for antibiotic use pertaining to the efficacy and knowledge of stewardship practices, and the knowledge of critical timepoints, for mastitis diagnosis and control.

## Figures and Tables

**Figure 1 animals-14-02332-f001:**
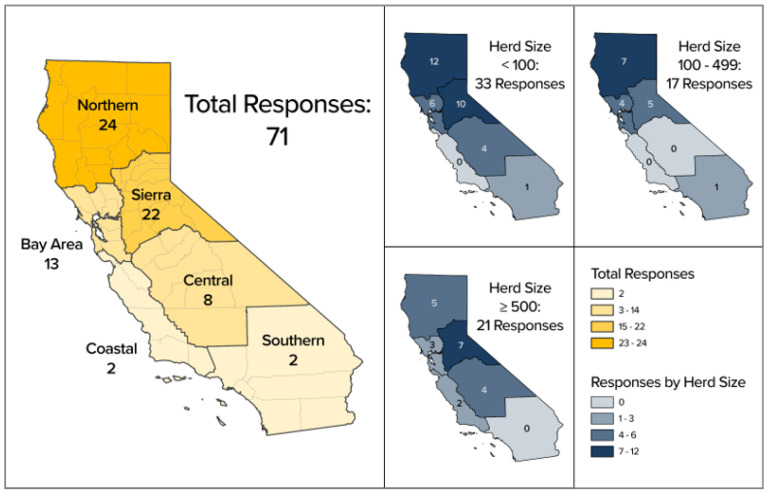
Distributions of survey responses by region in California and flock size.

**Figure 2 animals-14-02332-f002:**
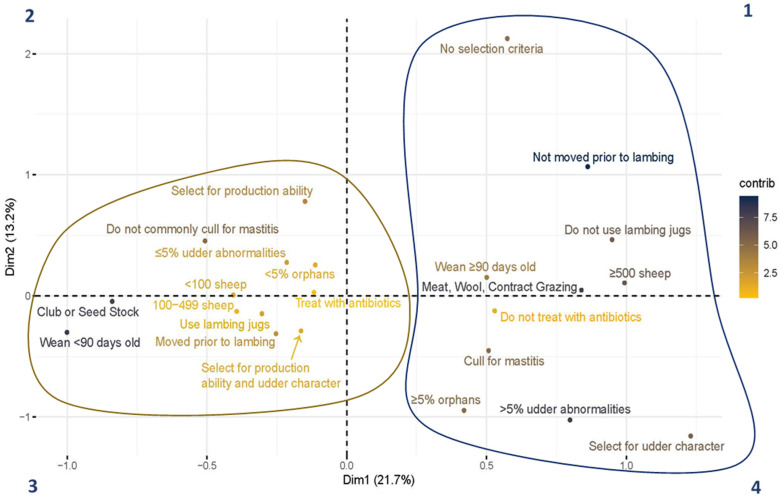
Multiple correspondence analysis of 10 flock-level variables. Quadrants are labeled by number. The color scale represents the contribution of the response to the variance in the dataset. Yellow represents low contribution and navy represents high contribution to variance.

**Table 1 animals-14-02332-t001:** Flock-level descriptive statistics with corresponding counts per category and row percentages.

Flock-Level Variable	Total (N)	Response Category	Count (n)	Percentage (%)
Flock Size	71	Fewer than 100	33	46%
		100–499	17	24%
		500 or greater	21	30%
Purpose ^	71	Meat, Wool, and/or Contract Grazing	36	51%
		Club/Seedstock	35	49%
Breeding Season	70	Winter (October–March)	20	29%
		Summer (from April to September)	50	71%
Tested for SRLV *	71	Yes	11	15%
		No	60	85%
Purchased Replacement Ewes	71	Yes	37	52%
		No	34	48%
Tested Replacement Ewes for SRLV *	37	Yes	2	5%
		No	35	95%
Used Udder Conformation in Selection Criteria	70	Yes	40	57%
		No	30	43%
Percentage of Udder Abnormalities Per Lactation	70	≤5%	54	77%
		>5%	16	23%
When Udder Abnormalities Were Noted ^	71	Pre-breeding	10	14%
		Mid-lactation	29	41%
		Lambing	47	66%
		Weaning	25	35%
		Post weaning	11	15%
Antibiotic Treatment Used for Mastitis	71	Injectable	55	77%
		Oral	2	3%
		Intramammary	26	37%
		None	4	6%
Ewes Moved to Close-up Area	70	Moved prior to lambing	54	77%
		Not moved for observation prior to lambing	16	23%
Used Lambing Jugs	70	Yes	53	76%
		No	17	24%
Weaning	68	<90 days old	23	34%
		≥90 days old	45	66%
Percentage of Orphan Lambs Per Season	69	<5% orphans	55	80%
		≥5% orphans	14	20%

* SRLV stands for small ruminant lentivirus; ^ indicates a multiselect response.

## Data Availability

The data presented in this study are available on request from the corresponding author.

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
