# Peer review of "Cross-Sectional Study Assessing Management Practices and Udder Health in California Sheep Flocks and Seroprevalence of Small Ruminant Lentivirus"

_animals, 2024, doi:10.3390/ani14162332_

Round 1

Reviewer 1 Report

Comments and Suggestions for Authors

Digianantonio et al. present a paper intitled “Cross-sectional study assessing management practices and udder health in California sheep flocks, and prevalence of OPP” which aimed to characterize the reproductive system of ovine in California as well as approach/determine the seroprevalence of OPP in the same region. For this effect the author used surveys sent online directly to producers of the region. First, we would like to congratulate the authors for the topic choice. We consider that including the producer perception of animals raisers on scientific research is very important. Also included a study of the prevalence of small ruminant lentivirus of this region, which caught all our attention.

So, we would like to concept a few comments and suggestions to the authors about the paper they wrote:

- Changing Ovine Progressive Pneumonia to Small Ruminant Lentiviruses. Knonw as Maedi-Visna in ovine species and Caprine Encephalitis Arthritis in caprine species, knowing nowadays that these viruses belong to the same group and depending on its subtype they can inflict disease in ovine, caprine or even both species. On our understanding, OPP represent just one of the clinical signs what may result from lentivirus infection.

- Abstract – well written with all the needed information to the paper understanding.

- Introduction – well written, succinct and approaching all the main topics. In our opinion some risk factors to SRLV infection should be described, adapted of course to the region production situation. In Europe many seroepidemiological studies have been made with the main aim to understand better the behavior of these viruses among herds, as well as comprehending its risk factors. Few examples of the studies that may be included in the introduction as well in discussion:

               - Seroprevalence and risk factors of exposure to caprine arthritis-encephalitis virus in southern Spain (Barrero-Domínguez et al. 2017)

               - Small Ruminant Lentivirus Infection in Sheep and Goats in North Portugal: Seroprevalence and Risk Factors (Jacob-Ferreira et al. 2023)

               - Small ruminant lentiviruses in goats in southern Italy: Serological evidence, risk factors and implementation of control programs (Cirone et al. 2019)

               - Bayesian estimation of seroprevalence of small ruminant lentiviruses in sheep from Poland (Olech et al. 2017)

               - Risk Factors Associated with the Alpine Multispecies Farming System in the Eradication of CAEV in South Tyrol, Italy (Tavella et al. 2021)

Material and methods: the use of digital platforms are methods that nowadays have to be considered and are important in order to reach more stakeholders more easily. However, we think that some questionnaires could have been carried out in person, carried out by a researcher or someone trained. Animal production still has elderly participants who may not work easily with these new resources. This way we thought they could have gotten more answers. Regarding the blood collection, we consider that this should have been carried out by a trained veterinary technician and not by a producer. In this way, we would adequately safeguard animal welfare and ensure that the blood comes from animals randomly selected for collection. Finally, we encourage the authors to continue and develop the study they started on SRLV.

Results and discussion: – well-presented and clear. We think they covered all the necessary aspects. We consider in relation to the SRLV that you could have mentioned whether your risk factors are in accordance with the bibliography, namely they were from your region, as we mentioned in the comment on introduction. 

Comments on the Quality of English Language

Line 8 – was the second (2022 is past)

Line 11/12 – hyphen needed “seventy-one” (because of the paragraph, but it is graphic adjustment)

Line 26 – “Chi-square”

Line 43 – Same as line 8, maybe start sentence with “In 2022, California was the second largest…”

Line 46 - “nursing”? or rather rearing? Nursing as a concept to human breast-feeding?

Line 48 – “rea-sons”?

Line 64 – “Thin” sheep?

Line 223-233: maybe delete all this paragraph?

Nothing to add to English revision besides some questions written above.

Author Response

Comments 1: Changing Ovine Progressive Pneumonia to Small Ruminant Lentiviruses. Known as Maedi-Visna in ovine species and Caprine Encephalitis Arthritis in caprine species, knowing nowadays that these viruses belong to the same group and depending on its subtype they can inflict disease in ovine, caprine or even both species. On our understanding, OPP represent just one of the clinical signs what may result from lentivirus infection.

Response 1: Thank you very much for your comment. While in the US all clinical manifestations of SRLV in sheep are commonly referred to as OPP, we agree that using this acronym may be misleading and have changed nearly every instance of OPP to SRLV in the manuscript. We did include one sentence to describe to US readers why we use SRLV throughout (lines 62-66).

Comments 2: Introduction – well written, succinct and approaching all the main topics. In our opinion some risk factors to SRLV infection should be described, adapted of course to the region production situation. In Europe many seroepidemiological studies have been made with the main aim to understand better the behavior of these viruses among herds, as well as comprehending its risk factors. Few examples of the studies that may be included in the introduction as well in discussion.

Response 2: Thank you again for this comment. We very much agree and added references to your point. In the US there are only the studies that are already referenced, so we further expanded the search and referenced  the European seroepidemiological studies you mentioned (lines 82-86).

Comments 3: Material and methods: the use of digital platforms are methods that nowadays have to be considered and are important in order to reach more stakeholders more easily. However, we think that some questionnaires could have been carried out in person, carried out by a researcher or someone trained. Animal production still has elderly participants who may not work easily with these new resources. This way we thought they could have gotten more answers.

Response 3: We agree with this statement. While we did recruit participants in person at events with producers of all ages, and one of the authors helped producers find the survey on their computers and phones during farm visits, the online survey format is a limitation to the study. We added the inperson recruitment to the methods section (lines 101-105).

Comment 4: Regarding the blood collection, we consider that this should have been carried out by a trained veterinary technician and not by a producer. In this way, we would adequately safeguard animal welfare and ensure that the blood comes from animals randomly selected for collection.

Response 4: Thank you for pointing out an important missing detail on our part. While one of the authors created sampling kits for producers to take home from in person meetings and ship back for testing, the sampling was carried out by their flock veterinarian or trained veterinary technicians. This was specifically highlighted in the manuscript. Thank you again. (lines 167-169)

Comments 5:  Finally, we encourage the authors to continue and develop the study they started on SRLV.

Response 5: Thank you very much for this encouraging comment. With little preliminary data it was difficult to find funding for a larger statewide effort. The authors are continuing research on SRLV transmission behavior in a research flock in CA, and with the publication of this manuscript, we have hopes to continue the SRLV work on a large scale.

Comment 6: Results and discussion: – well-presented and clear. We think they covered all the necessary aspects. We consider in relation to the SRLV that you could have mentioned whether your risk factors are in accordance with the bibliography, namely they were from your region, as we mentioned in the comment on introduction. 

Response 6: Additional references were added to the discussion to support the findings from this study and how they relate to studies performed in other regions.

Reviewer 2 Report

Comments and Suggestions for Authors

Dear authors

This is a well written manuscript with minimal requirements for improvements.  I hope all my comments are taken only as a constructive criticism only.

General comment

In the materials and methods and results, you mention data were collapsed, as appropriate.  As appropriate is a very vague statement.  It would be better in the statistical analysis section to briefly explain what data we re-categorized, and how (e.g., include a data manipulation subsection of paragraph)

Specific comments

Please use the term 'flock' rather than interchangeably use 'flock' or 'herd' (throughout text or tables, please)

L253 'Lamb processing' please include in brackets what is typically included , as this may vary between regions and countries.

Please check for typos (e.g., L31 ex-tensive, L48 rea-sons, L163 an extra 's' before commercially, L170 space missing at the start of the sentence, L199 extra space at the start of the sentence, L 289 'indicate by' should be 'indicated in', L375 and L394 missing a comma, etc.)

Author Response

Comments 1: In the materials and methods and results, you mention data were collapsed, as appropriate.  As appropriate is a very vague statement.  It would be better in the statistical analysis section to briefly explain what data we re-categorized, and how (e.g., include a data manipulation subsection of paragraph).

Response 1: Thank you very much for your time and dedication. Your review is very much appreciated. Please find the addition of an example of the consolidation of variables from lines 130 to 136.  As this was performed for many questions it would be difficult to report on every instance. I hope this addition helps to explain how we consolidated variables. Thank you.

Comment 2: Please use the term 'flock' rather than interchangeably use 'flock' or 'herd' (throughout text or tables, please).

Response 2: Thank you very much for noticing this error. The changes were made throughout.

Comment 3: L253 'Lamb processing' please include in brackets what is typically included , as this may vary between regions and countries.

Response 3: Great point. A paranthetical was added in line 265 Lamb processing (i.e. ear tag placement, castration, and tail docking) 

Comment 4: Please check for typos (e.g., L31 ex-tensive, L48 rea-sons, L163 an extra 's' before commercially, L170 space missing at the start of the sentence, L199 extra space at the start of the sentence, L 289 'indicate by' should be 'indicated in', L375 and L394 missing a comma, etc.)

Response 4: Thank you so much for finding these errors. They have been addressed throughout.

Reviewer 3 Report

Comments and Suggestions for Authors

Dear Authors,

I have reviewed your manuscript titled "Cross-sectional study assessing management practices and udder health in California sheep flocks, and prevalence of OPP." 

Suggestions for Improvement:

  1. Response Rate and Sampling Bias: The response rate of 19.8% is relatively low, and the overrepresentation of larger flocks may lead to sampling bias. Consider discussing these limitations more explicitly and suggest strategies to mitigate them in future studies.
  2. Given the overrepresentation of larger flocks, your findings may not be fully generalizable to all sheep flocks in California. This should be clearly stated in the limitations section..
  1. Typographical Errors:
    • On page 2, line 28: "was were" should be corrected to "were".
    • On page 3, line 48: "rea-sons" should be corrected to "reasons".
    • On page 11, line 382: "base" should be corrected to "based".
    • Ensure consistent use of terms like "ewe-level prevalence" and "flock-level prevalence" throughout the manuscript.
  2. Clarity and Readability:

    • The sentence on page 12, lines 321-325, could be broken down into shorter, more concise sentences.

Overall, your manuscript is well-structured and presents significant findings that contribute to the understanding of sheep health management. With the suggested improvements, your study will provide even more robust and ready to be published

Sincerely,

Author Response

Comments 1: Response Rate and Sampling Bias: The response rate of 19.8% is relatively low, and the overrepresentation of larger flocks may lead to sampling bias. Consider discussing these limitations more explicitly and suggest strategies to mitigate them in future studies.

Response 1: Thank you very much for your time and expertise in providing a review of the manuscript. Clarification was added to the discussion of sampling bias and the low response rate from lines 345 to 350.

Comments 2: Given the overrepresentation of larger flocks, your findings may not be fully generalizable to all sheep flocks in California. This should be clearly stated in the limitations section.

Response 2: Thank you for this comment. "The over-representation of larger flocks may indicate that the results are not generalizable to California sheep flocks. This could have been limited by performing outreach for survey participation to more diverse groups (i.e. 4H groups and small farm flock producers producers)." lines 363-366. 

Comments 3: typographical errors

  • On page 2, line 28: "was were" should be corrected to "were".
  • On page 3, line 48: "rea-sons" should be corrected to "reasons".
  • On page 11, line 382: "base" should be corrected to "based"

Response 3: Thank you for catching these errors. They have been addressed.

Comment 4: Ensure consistent use of terms like "ewe-level prevalence" and "flock-level prevalence" throughout the manuscript.

Response 4: These have been adjusted throughout.

Comment 5: The sentence on page 12, lines 321-325, could be broken down into shorter, more concise sentences.

Response 5: I am having a difficult time identifying the specific sentence referred to by this comment, likely due to formatting differences. I hope were adequately addressed the issue.

Reviewer 4 Report

Comments and Suggestions for Authors

This is an interesting study performed in sheep flocks in California, USA. The authors have collected a substantial amount of data from an extremely large number of flocks in the area. This was achieved through regular visits to the flocks and by means of a rigorous and very detailed questionnaire. This is by far one of the largest studies ever carried out worldwide and the authors must be commented on for such an excellent work.

The manuscript is almost ready for acceptance. I only have a few minor points that require clarification and which will help to improve the final document.

1.      Please make clear the advantages of this work over similar studies performed previously on an international basis.

2.      Please explain the gaps in the literature that will be filled after publication of your excellent study.

3.      Please clarify the means through which the visits to the farms were made: private vehicle, corporate vehicle, bicycle etc.

4.      Please explain what type of control procedures you implemented during the study; for example, did you use control questionnaires to prove the correctness of answers? did you use farm in other US states to compare results? This will be helpful for the evaluation of the findings.

5.      I suggest to carry out a similar analysis for OPP and show the results, but I leave that to the authors, as perhaps they may wish to make a second paper with these findings.

6.      The text in the Results section can be significantly reduced by adding tables summarizing the results. Such a change will facilitate the flow of reading.

7.      Please separate the Discussion section into sub-sections, as the Discussion covers various topics with limited relevance between them.

8.      I suggest to increase graphs in the manuscript, in order to improve visualization of the manuscript.

9.      The references are OK.

10.  The concluding section is OK.

Overall. An excellent manuscript that requires only limited changes before final acceptance.

Author Response

Comments 1: Please make clear the advantages of this work over similar studies performed previously on an international basis.

Response 1: Thank you very much for your time and expertise you committed to reviewing this manuscript. We added a line to address your important point (lines 84-86).

Comment 2: Please explain the gaps in the literature that will be filled after publication of your excellent study.

Response 2: Thank you for this comment. We added references to international literature and explained the importance of this study in the introduction (lines 82-86).

Comment 3:  Please explain what type of control procedures you implemented during the study; for example, did you use control questionnaires to prove the correctness of answers? did you use farm in other US states to compare results? This will be helpful for the evaluation of the findings.

Response 3: There were no control procedures for this study. 

Comment 4: I suggest to carry out a similar analysis for OPP and show the results, but I leave that to the authors, as perhaps they may wish to make a second paper with these findings.

Response 4: The authors attempted to do this analysis but we did not think the results were strong enough to report with too small of a sample size.

Comments 5: Please separate the Discussion section into sub-sections, as the Discussion covers various topics with limited relevance between them

Response 5: Thank you for this comment. We have made this change.

Round 2

Reviewer 4 Report

Comments and Suggestions for Authors

All the issues raised were covered. No further comments.